

# Thinking about it: the impact of COVID-19-related stimuli on prospective memory

Stefano Vicentin[1,2], Giovanni Cantarella[3], Giorgia Cona[1,2] and Patrizia Bisiacchi[1,2]

[1] Department of General Psychology, University of Padua, Padua, Italy
[2] Padua Neuroscience Center, University of Padua, Padua, Italy
[3] Department of Psychology, University of Bologna, Bologna, Italy

## ABSTRACT

**Background**. Since 2020, information regarding COVID-19 has been a constant presence in the news, in our conversations and thoughts. Continuous exposure to this type of stimuli could have an impact on cognitive processes essential for our everyday activities, such as prospective memory (PM). PM is the ability to remember to perform an intention at a specific point in the future, like remembering to take prescribed medicines at a specific time or to turn off the stove after cooking. Do COVID-related stimuli affect our ability to perform a PM task?

**Methods**. To answer this question, we proposed a novel version of the classical paradigm used to investigate PM. Namely, this paradigm includes a baseline condition, in which an ongoing task is presented alone, and a PM condition in which the same task is proposed again together with a second (prospective) task. In this study, a short video clip was presented between the baseline and the PM condition. The video clip displayed either neutral, negative, or COVID-related content. Additionally, participants were asked to respond to two questionnaires and a series of questions regarding their well-being and experience with the pandemic. Namely, the DASS-21 scale (evaluating depression, anxiety, and stress), and the COVID-19-PTSD questionnaire (a questionnaire evaluating post-traumatic stress disorder symptoms related to the pandemic experience) were administered. Participants' performance and responses were analyzed using a linear mixed effect (LME) model approach, and correlation analyses were run to highlight possible correlations between participants' scores in the DASS-21, the COVID-19-PTSD, and the additional questions on their personal experience with the pandemic.

**Results**. The LME models revealed significant effects of the displayed video on performance: in line with previous studies, the clip displaying standard negative contents led to impaired accuracy in the ongoing task in the PM condition, compared to the Baseline. In contrast, participants who saw the COVID-related clip showed improved accuracy in the ongoing task compared to the other participants, selectively in the block performed after the video clip was displayed (PM condition). Furthermore, the explanatory power of the LME model calculated on accuracy to the ongoing trials was enhanced by the inclusion of the scores in the anxiety subscale of the DASS-21, suggesting a detrimental role of anxiety. Altogether, these results indicate a different effect of the exposure to classical negative contents (associated with a cost in terms of accuracy in the ongoing task between the baseline and the PM condition) and the

Corresponding author
Stefano Vicentin,
stefano.vicentin@unipd.it

pandemic-related one, which was instead characterized by a higher accuracy to ongoing trials compared to the other video clips. This counterintuitive finding seems to suggest that COVID-related stimuli are processed as "acute stressors" rather than negative stimuli, thus inducing a state of increased alertness and responsivity.

# INTRODUCTION

Portions of the text presented here were previously published as part of a preprint (*Vicentin et al., 2023*). At the end of 2019, a new type of coronavirus, the SARS-CoV-2 (or COVID-19) began to spread worldwide, causing the largest pandemic of the 21st century. Since then, information regarding COVID-19 has been a constant and overwhelming presence in our daily lives, changing people's priorities and affecting their feelings, relationships, and thoughts. The pandemic outbreak of an unrecognized infection with no effective medical treatments was defined as a traumatic experience due to its acute and chronic implications at individual and community levels (*Brooks et al., 2020*). The *World Health Organization (2020)* reports that the pandemic has induced a considerable degree of fear, worry, and concern in the general population, resulting in an overall worsening of individuals' psychological health. In the most affected regions, such as China, India, and Italy, studies reported a higher severity and prevalence of depression and anxiety disorders, together with symptoms such as abnormal sleep, appetite changes, health anxiety and reduced libido, both during the lockdown and one year after its beginning (*Fiorenzato et al., 2021*; *Fiorenzato & Cona, 2022*; *Saraswathi et al., 2020*; *Wang & Zhao, 2020*; *Zhang et al., 2020*). Furthermore, another study highlighted the development of post-traumatic stress disorder (PTSD) symptoms in the general population, independently from the contraction of the COVID-19 virus (*Sun et al., 2021*).

Which specific factor could have affected people's mental health during the pandemic? Researchers found that one of the main worries was the fear of contagion—that is, concern over being infected by the virus, also linked to the risk of death for oneself and loved ones. Other detrimental factors to personal well-being were loneliness, the fear of job loss, and the absence of interpersonal relationships following the physical and social distancing imposed during the lock-down and quarantine periods (*Fiorenzato & Cona, 2022*; *Kmietowicz, 2020*; *Xiao et al., 2020*). Such feelings were associated with the severity of anxiety symptoms determined by the high contagiousness and rapid spread of COVID-19 (*Huang & Zhao, 2020*; *Mazzuca et al., 2022*).

Negative affective states such as stress and anxious feelings are known in the literature to affect cognitive processes (*Carboni et al., 2017*; *Guez et al., 2016*; *Tiferet-Dweck et al., 2016*; *Vromen et al., 2016*). To investigate the impact of affective states on cognitive functions, researchers often rely on the affective priming paradigm (*Klauer & Musch, 2003*). Briefly, this paradigm involves the presentation of salient, affective-charged stimuli before the

presentation of a specific task, in which the process of interest is performed. The results are compared with the ones of the same task presented in different conditions, either not preceded by an affective prime, or preceded by primes with different emotional charges. The strength of the influence that affective priming can have on cognitive performance can be modulated by two features of the emotional stimulus, identified as the most basic dimensions of emotional information: valence and arousal (*Imbir, Pastwa & Walkowiak, 2021*; *Russell, 2003*; *Zhang, Kong & Jiang, 2012*). Valence is defined as the level of pleasantness that is generated when a stimulus is encoded, which varies from negative (displeasure) to positive (pleasure). Arousal refers to the degree of physiological activation that a stimulus elicits, varying from calm (low) to excited (high) (*Lang, Bradley & Cuthbert, 1997*). Several studies have shown that using stimuli with a positive valence to induce affective priming results in a facilitatory effect on performance, whereas stimuli with a negative valence display inhibitory roles (*Yao & Wang, 2013*; *Sass et al., 2012*; *Kissler & Koessler, 2010*; *Rossell & Nobre, 2004*). That being said, several studies suggest that it's the levels of arousal that the emotional stimuli elicit, rather than their valence, to have a greater impact on perception and memory (*e.g.*, *Kensinger, 2009*; *Mather & Sutherland, 2011*; c.f. *Adelman & Estes, 2013*).

Different types of emotional stimuli have been validated and used to modulate cognitive performance, from written words to images or sounds (*Bradley, Cuthbert & Lang, 1998*; *Bradley & Lang, 1999*; *Lang, Bradley & Cuthbert, 1998*). In literature, the most used affective stimuli are pictures, as they are easy to implement in an experiment, can be proposed several times, and provide the experimenter with the flexibility to switch between different presentation paradigms. Recently, more dynamic and long-lasting stimuli such as video clips and music have also been employed, in the attempt to obtain more ecological and spontaneous elicitation of the emotion of interest (*Maffei & Angrilli, 2019*; *Zentner, Grandjean & Scherer, 2008*). Video clips, in particular, show a great potential for eliciting emotional responses (*Romeo et al., 2022*; *Westermann et al., 1996*).

A meta-analysis conducted by *Murphy & Isaacowitz (2008)* demonstrated that the modulation of valence and arousal levels of the emotional stimuli can produce a great effect on several cognitive processes, such as attention and memory. An ability strictly connected with these two functions, but less studied from the affective priming perspective, is prospective memory (PM). PM is the ability to remember to perform a specific intention at the occurrence of a stimulus previously associated with it (*Brandimonte, Einstein & McDaniel, 1996*). It is a fundamental ability for our everyday activities, from remembering to stop at the grocery store after work to remembering to take medication once a day at a specific time. This function is considered a multi-faceted mechanism, as many cognitive processes are involved in the successful execution of PM. First, the intention must be encoded in memory and associated with a stimulus (encoding phase) that will work as a trigger to retrieve the intention at the right time. While the individual monitors the environment to check for the occurrence of that trigger (the so-called *PM cue*), they must simultaneously maintain the intention in their mind and engage in other activities (maintenance phase). Finally, when the PM cue appears, they have to recognize it and recall the associated intention from memory (retrieval phase).

Hence, PM requires the balancing of resources between mnestic processes, to maintain the intention in memory, and attentional ones, to monitor the environment for the occurrence of the conditions associated with its retrieval. These processes work in parallel, competing cognitive resources that are, by nature, limited. According to the Multiprocess Framework (*McDaniel & Einstein, 2000*; *McDaniel & Einstein, 2007*; *Scullin, McDaniel & Shelton, 2013*), the intention can be successfully retrieved in two different ways. The first mechanism, defined as strategic monitoring, consists of the continuous monitoring of the environment, searching for the occurrence of the PM cue (*i.e.,* target checking) while actively maintaining the intention in memory (*i.e.,* retrieval mode) (*McDaniel & Einstein, 2000*; *Guynn, 2003*; *Cona et al., 2012*). Under different circumstances, a second process called spontaneous retrieval can be used to automatically recall the intention, without the allocation of resources to monitor the environment (*Moscovitch, 1994*; *McDaniel & Einstein, 2000*). Multiple factors modulate the preferential activation of either strategic monitoring or spontaneous retrieval processes to perform a PM task. The variables that most consistently showed to impact this mechanism selection are the focality and the perceptual salience of the PM cue (*Cona, Bisiacchi & Moscovitch, 2014*; *Cona et al., 2016*). Focality refers to the degree of overlap between the ongoing task (the activity participants are involved in while maintaining the intention) and the PM task, *i.e.,* the action that must be performed at the presentation of the PM cue. Salience, on the other hand, refers to the degree of perceptual similarity between the PM cue and the other ongoing trials—that is, how outstanding the PM cues are among the ongoing stimuli. Several studies showed that salient and focal PM cues are easier to detect and prompt the reliance on more automatic processes (*i.e.,* spontaneous retrieval), whereas non-focal and non-salient cues require more attentional resources to be detected, thus encouraging strategic monitoring (*Einstein et al., 2005*; *Scullin et al., 2010*). Critically, increased monitoring requests to perform the PM task reduces the resources available to perform the ongoing task, thus worsening its performance. The difference in terms of reduced performance to the ongoing task between a condition in which the task is presented alone (baseline condition) and a condition in which the same task is proposed together with some prospective instructions (PM condition) is known in the literature as the PM cost (*Hicks, Marsh & Cook, 2005*). In line with the Multiprocess Framework, the PM cost can be considered an indicator of the selected mechanism, since performing a PM task through strategic monitoring is reflected in great PM costs in the ongoing task, whereas spontaneous retrieval is characterized by the absence of this phenomenon (see Anderson, 2019 for a review). Thus, performance in the ongoing task is considered a valuable tool for the investigation of the underlying PM processing.

The emotional valence of the PM cues is another factor that has been shown to affect PM. In this regard, *Clark-Foos et al. (2009)* found that both pleasant and unpleasant PM cues were detected less frequently than neutral ones, suggesting that valence has the ability to usurp attentional resources that would have otherwise supported successful PM performance. Negative stimuli, however, resulted in a greater impairment in PM cue detection. Instead, *Rendell et al. (2011)* found that pleasant PM cues were associated with a better performance in the PM task, thus revealing a positivity-related enhancement, but
no differences depending on negative stimuli. More recently, *Cona et al. (2015)* performed a study to disentangle the effects of the emotional valence on PM cues on the distinct PM phases—namely, encoding, maintenance, and retrieval phases. The emotional valence of both the ongoing stimuli and the PM cues was manipulated (pleasant, neutral, unpleasant). Emotional PM cues (both pleasant and unpleasant) not only enhanced the automatic, bottom-up capture of attention (spontaneous retrieval) but also increased the activation of top-down processes (strategic monitoring), improving the detection of the PM cue and the retrieval of the intention from memory. All in all, the empirical findings on the effect of emotions on PM can be considered still sparse and inconclusive, compared to those regarding other cognitive functions. However, the importance of PM processes in several everyday activities highlights the importance of investigating the relationships between PM and emotions more deeply.

Since 2019, people's emotional and psychological well-being has been challenged by the outbreak of the COVID-19 virus and the continuous exposure to information regarding its development and victims (*Hamidein, Hatami & Rezapour, 2020*; *Pedrosa et al., 2020*). In the years that followed, the impact of the pandemic on mental health was investigated mostly through the employment of self-report questionnaires and surveys (*Cannito et al., 2020*). Despite their utility in highlighting the difficulties and the impairment in everyday functioning, however, these methodologies lack the quantitative properties to evaluate the extent of the impact of the pandemic's aftermath on different cognitive processes. As the literature has demonstrated, the implementation of an affective priming paradigm may shed light on the impact of emotions on such cognitive functions (*Murphy & Isaacowitz, 2008*; *Rokke & Lystad, 2015*; *Chen et al., 2008*). Therefore, implementing this paradigm to investigate the impact of stimuli associated with the pandemic situation can help to understand the impact of the related emotions on essential everyday abilities. In light of this theoretical background, in the present study, we explore the effects of COVID-related stimuli on PM, to investigate the impact of the currently experienced situation on this essential process for our everyday functioning. For this aim, we first validated a series of COVID-related video clips, to select one as a COVID-related affective prime. Subsequently, two video clips (one displaying negative and the other neutral contents) with similar properties were chosen from a recently validated dataset (*Maffei & Angrilli, 2019*). These three video clips were inserted in a classical PM paradigm, immediately after the encoding phase of the PM task. Participants were randomly assigned to one of the three versions of the paradigm, differing only for the presented video clip (COVID-related, negative, or neutral). In addition, two questionnaires were administered to each participant, one exploring their depressive, state-anxiety, and stress levels (DASS-21 scale; *Bottesi et al., 2015*), the other investigating the presence of PTSD symptoms related to the pandemic experience (COVID-19-PTSD questionnaire; *Forte et al., 2020*). Finally, questions regarding their personal experience with the pandemic were addressed to explore the presence of possible correlations between PM cue detection, PM costs, the responses to the questionnaires, and the impact of the pandemic on each participant's everyday activity.
## MATERIALS & METHODS

### Participants

Data were acquired from a total of 339 volunteers recruited through Facebook, placing announcements in university-related groups. Among them, 27 participants were excluded because their age was above the range of interest (18–38). Furthermore, 16 did not complete the final part of the questionnaire and 13 did not respond to at least 60% of the presented trials in one of the experimental conditions. Additionally, the performance of five participants classified them as outliers (accuracy to ongoing trials was lower than 2.5 standard deviations from the mean), leading to the final population of 278 participants ($M = 24.1$, range $= 18$–$38$, 86 males). The study followed the guidelines of the Helsinki declaration and was approved by the ethics committee of the University of Padova (Protocol number: 3931). Every participant expressed their written informed consent before the beginning of the experiment.

### Stimuli and materials

The paradigm was composed of three main elements: a questionnaire, a video clip, and the proper experimental task. The questionnaire consisted of two different parts, one presented before the experimental task and the other after, as discussed in the Procedure section. The first part of the questionnaire consisted of general demographic questions (age, gender, title of studies) and the DASS-21 scale, a self-report questionnaire investigating the levels of stress, depression, and anxiety participants experienced in the previous seven days (*Bottesi et al., 2015*). The second part of the questionnaire, proposed at the end of the experimental task, consisted of three subscales of the COVID-19-PTSD questionnaire, and questions regarding the personal experience with the pandemic. The COVID-19-PTSD questionnaire is a recently validated scale to evaluate the presence of symptoms of post-traumatic stress disorder related to the pandemic experience in the general population (*Forte et al., 2020*). Specifically, the Intrusion, Avoidance, and Dysphoric Arousal subscales were selected to investigate the possible correlation between the scores in these domains and the effect of the contents of the videos on performance. To collect further information regarding each participant's personal experience with the pandemic, in this second part of the questionnaire we asked them if they had ever contracted the virus, whether they were positive at that moment, how much they feared the infection, and how often they searched for information regarding the pandemic situation. Additionally, we also asked them to judge the impact of COVID-19 on several domains of their everyday living (occupation, social life, family, relationships, psychological and physical health), and about their intention to require psychological support to cope with the pandemic's aftermath.

The experimental paradigm consisted of a classical PM paradigm, composed of a baseline and a PM condition. In the baseline condition, participants were required to perform the lexical decision task (LDT) without any additional (prospective) request. The LDT is a classical task used in research on PM, that requires participants to judge as quickly and accurately as possible the lexicality of different strings of letters. In the following PM condition, participants were asked to continue performing the LDT (serving as the *ongoing task*) but also to simultaneously perform a *PM task*, consisting of the request to

press a specific key whenever a specific syllable ("MA") appeared (corresponding to the intention to maintain in memory). For our research, we selected a non-focal, non-salient PM task since this type of prospective instruction is known in the literature to increase the monitoring requests of the PM process (*Einstein et al., 2005*; *Scullin et al., 2010*), promoting the reliance on strategic monitoring and inducing greater PM costs, thus increasing the variability in participants' responses (and, subsequently, the chances to detect a difference associated with the affective prime).

Both the baseline and the PM conditions consisted of 100 different strings of letters (half words, and half pseudowords) matched between conditions for psycholinguistic variables such as frequency, length, and initial letter. In the PM condition, eight additional trials containing the target syllable (the so-called *PM cue*) were placed within the other trials (frequency of the PM cue: 7,4%), leaving at least 10 trials between each PM cue. The strings of letters were selected from a dataset used in a previous study to investigate PM processing (*Vicentin et al., 2022*).

The experiment was programmed with the free software PsychoPy (version 2020.1.2; *Peirce, 2007*; *Peirce et al., 2019*). Each trial was structured in the same way: first, a fixation symbol (+) appeared in the center of the screen for a pseudorandom duration (1,000, 1,250, or 1,500 ms), to prepare the participant for the presentation of the item. After this period, a string of white letters was presented in the center of a gray screen, using the font Lucida Console. The stimuli lasted until one of the possible keys was pressed or for 2,000 ms at most. After this phase, a blank screen appeared for 1,000 ms, serving as an interstimulus interval. The order of presentation of the trials was determined through an online true random generator, RANDOM.org (*Haahr, 2020*).

The last component of the paradigm was a brief video clip displayed between the baseline and the PM conditions. Participants were presented with one of three possible movie excerpts, two of which were part of the E-MOVIE dataset, a collection of emotional stimuli validated by *Maffei & Angrilli (2019)*. Specifically, the first video was selected from the category "Neutral" and consisted of an excerpt from a documentary on a small city in Italy, *Calamonaci*. The version of the paradigm presenting this neutral prime was defined as the DOC (Documentary) condition. The second video was selected from the category "Compassion" and consisted of an excerpt from the movie *The Pursuit of Happiness*. This video was used to form the version of the paradigm with negative valence, labeled as the NEGA condition. Finally, the third video clip was an excerpt from the movie *Contagion* (*Soderbergh, 2011*), a movie whose contents refer to a pandemic situation strongly reminding COVID-19. The movie excerpt for this category was selected among six different video clips, all containing strong references to the pandemic situation, that we previously validated using the same procedure applied for the validation of the E-MOVIE dataset (see 'Video Validation' section below). Among the others, we selected the clip labeled as the *Cough* excerpt, as it presented valence and arousal values similar to the ones of the video selected for the NEGA condition (see Table 1). The version of the paradigm presenting this COVID-related video clip constituted the third version of the experiment, labeled as the COV condition. The three movie excerpts can be found as Supplemental Information.

**Table 1** Valence and arousal levels of the video clips selected as affective primes for the experiment.

| Video clip | Category | Valence | Arousal |
|---|---|---|---|
| Cough | Covid (COV) | $F = 2.64 \pm 1.48$ | $F = 5.56 \pm 2.08$ |
| | | $M = 2.71 \pm 1.61$ | $M = 5.41 \pm 1.87$ |
| The Pursuit of Happiness | Negative (NEGA) | $F = 3.33 \pm 2.11$ | $F = 5.62 \pm 2.28$ |
| | | $M = 3.26 \pm 1.58$ | $M = 5.06 \pm 2.11$ |
| Italian Documentary: Calamonaci | Neutral (DOC) | $F = 4.7 \pm 1.26$ | $F = 2.00 \pm 1.62$ |
| | | $M = 4.56 \pm 1.82$ | $M = 2.23 \pm 1.75$ |

Notes.

Valence and arousal levels for each of the selected videos, divided by gender (F, Females, M, Males). The values for the clip "Cough" were obtained through the validation explained below. The values of the clips "The Pursuit of Happiness" and "Italian Documentary: Calamonaci" were obtained from the E-MOVIE dataset.

## Video validation

Starting from different documentaries on the pandemic and from the movie *Contagion* (a movie containing strong parallelisms with the pandemic situation), we created six different video clips. In the realization of the stimuli, we considered the following criteria: video clips had to have the same duration as the ones in the E-MOVIE dataset (two minutes), to be in high definition (full HD), to present a coherent narrative without abrupt transitions, and no texts or subtitles could be present. This requirement reduced the possible sources of distraction and allowed a more direct comparison with the movie excerpts selected for the negative and neutral conditions. Each video was structured to present the most salient event in the second half of the scene, thus creating a gradual increase in emotional response during the narration. A total of 56 healthy volunteers ($M = 21.9$, range 18–26, 17 males) took part in the validation of the stimuli. The validation procedure consisted of the presentation of a total of eight videos (six COVID-related video clips, and two neutral video clips), each followed by a questionnaire in which participants were required to judge the video clip they just saw on different emotional domains. Before the beginning of the experiment, each participant expressed informed consent. Participants performed the validation online in a single session, lasting approximately 30 min. Participants were addressed to one of four possible links, each one containing all the videos, but presented in four different orders. Independently from the order, the two neutral videos of the E-MOVIE dataset were presented in the first and the fifth position, the former to allow participants to familiarize with the instructions and the questions of the survey, the latter to partially compensate for the effects of the cumulative presentation of COVID-related video clips.

As mentioned above, an assessment was proposed after each video clip, using a questionnaire composed of the same questions used in the validation of the E-MOVIE dataset: We asked participants to indicate whether they had already seen the clip (*Familiarity score*), how aesthetically beautiful they considered it, and if during the presentation they had closed their eyes or looked away. Similar to the validation of the E-MOVIE dataset, the following section of the assessment included a digital version of the *Self-Assessment Manikin* (*Bradley & Lang, 1994*), in which we asked participants to rate the pleasantness of the emotional state elicited by the clip and the arousal felt during the vision. Consequently,

**Table 2  Valence and arousal levels of the validated video clips.**

| Video clip | Category | Valence | Arousal | Covid |
|---|---|---|---|---|
| Bronte | Neutral | $F = 5.82 \pm 1.70$ | $F = 3.28 \pm 1.84$ | $F = 1.87 \pm 1.64$ |
| | | $M = 5.71 \pm 1.40$ | $M = 3.65 \pm 2.40$ | $M = 1.41 \pm 1.28$ |
| Cough | Covid | $F = 2.64 \pm 1.48$ | $F = 5.56 \pm 2.08$ | $F = 5.95 \pm 1.50$ |
| | | $M = 2.71 \pm 1.61$ | $M = 5.41 \pm 1.87$ | $M = 5.82 \pm 1.07$ |
| No_tomorrow | Covid | $F = 2.15 \pm 1.48$ | $F = 5.80 \pm 2.26$ | $F = 6.87 \pm 0.52$ |
| | | $M = 2.59 \pm 1.66$ | $M = 5.41 \pm 2.21$ | $M = 6.77 \pm 0.66$ |
| Bat | Covid | $F = 2.49 \pm 1.78$ | $F = 5.82 \pm 2.19$ | $F = 5.69 \pm 1.57$ |
| | | $M = 2.94 \pm 2.11$ | $M = 5.59 \pm 1.81$ | $M = 5.12 \pm 1.76$ |
| Quartesolo | Neutral | $F = 5.74 \pm 1.53$ | $F = 3.15 \pm 2.10$ | $F = 1.54 \pm 1.10$ |
| | | $M = 5.06 \pm 1.44$ | $M = 2.59 \pm 1.42$ | $M = 1.12 \pm 0.49$ |
| Hospital | Covid | $F = 1.80 \pm 1.51$ | $F = 6.15 \pm 2.39$ | $F = 6.87 \pm 0.41$ |
| | | $M = 1.82 \pm 0.88$ | $M = 6.00 \pm 2.45$ | $M = 6.88 \pm 0.49$ |
| Father_Daughter | Covid | $F = 3.46 \pm 1.95$ | $F = 5.03 \pm 1.99$ | $F = 5.90 \pm 1.14$ |
| | | $M = 4.41 \pm 1.70$ | $M = 4.24 \pm 1.64$ | $M = 4.59 \pm 1.58$ |
| Negationist | Covid | $F = 2.08 \pm 1.48$ | $F = 5.64 \pm 2.25$ | $F = 6.64 \pm 1.11$ |
| | | $M = 1.77 \pm 0.66$ | $M = 5.64 \pm 1.84$ | $M = 6.82 \pm 0.53$ |

Notes.

Valence and arousal levels for each of the eight validated video clips. Apart from the first and the fifth videos, which belong to the Neutral category, each of the other six videos was compared with the ones in the "Compassion" category of the E-Movie dataset. In the selection, we also considered the score at the "Covid-relatedness" scale.

participants had to evaluate the emotions felt during the vision of the video, rating the level of activation of six basic emotions (Fear, Sadness, Rage, Disgust, Joy, and Surprise) on a Likert scale (from 0 to 9). The assessment continued with an evaluation of the clip composed of a list of 15 emotional adjectives, selected to encompass a broad span of positive and negative feelings. Participants were asked to indicate on a 5-point Likert scale how much they felt in the way suggested by the emotional adjective. Finally, participants judged, on a 7-point Likert scale, to what extent the clip made them think about the COVID-19 pandemic situation. For the validation, we compared the six video clips based on these multiple dimensions (primarily valence and arousal) with the negative E-MOVIE excerpts. Among the six validated video clips and the negative stimuli of the E-MOVIE dataset, the COVID-related video clip *Cough* and the movie excerpt from *The Pursuit of Happiness* presented the most similar values of valence and arousal. The results of Independent Samples T-Tests confirmed the absence of significant differences between the valence (males: $t.\text{value} = -1.42$, $p = .175$; females: $t.\text{value} = -1.64$, $p = .108$) and arousal (males: $t.\text{value} = 0.77$, $p = .450$; females: $t.\text{value} = -0.1$, $p = .867$) scores of the two video clips. Additionally, the *Cough* video clip elicited a strong level of induction for COVID-related thoughts (5.9 out of 7). Therefore, these two video clips were selected as the affective primes for the COV and the NEGA conditions. Table 2 displays an overview of the values of the validated video clips.

## Procedure

Participants were recruited online using university-related Facebook groups, between September 2021 and February 2022. Every participant received a link to a Qualtrics page (Qualtrics, Provo, UT, USA), where they expressed their informed consent. Subsequently, the first part of the survey was displayed. This first part was composed of demographic questions and the DASS-21 scale. This scale was proposed here to collect information on these dimensions before the beginning of the experimental task and, therefore, independently from the effects of the affective priming stimulus (*i.e.,* the video clip). Once participants completed this first phase, they were presented with a link to the experimental paradigm, administered through Pavlovia.org (*Peirce et al., 2019*), an online platform where PsychoPy-built experiments can be conducted. First, instructions to the LDT were given. After that, participants performed a small practice phase (10 trials) to become familiar with the task. After this phase, they performed the first condition of the paradigm, the baseline (LDT only). Once this first part was finished, they received the instructions for the PM condition (composed of the LDT and the PM task) and were warned that, right before the beginning of the task, a short video would be displayed, to which they were required to pay attention. Consequently, one of the three different videos (neutral, negative, and COVID-related) was shown, immediately followed by the PM condition. Each version of the paradigm (and, therefore, each different video clip), was randomly presented to one-third of the participants (COV: 93; NEGA: 93; DOC:92). At the end of this second experimental block, three questions were asked about the contents of the video clip, to make sure participants paid attention and maintained the contents of the video in memory while performing the task (*e.g.,* "Were there scenes where some character was on the phone?"). All participants composing the final population responded correctly to at least two of the three presented questions, thus confirming that they had paid attention and maintained the information in memory while performing the PM condition.

Once the experiment was finished, participants were re-addressed to the Qualtrics page, to complete the final part of the survey. This last section was composed of the COVID-19-PTSD questionnaire and questions on the impact that the pandemic had on several domains of participants' everyday life (social life, family, relationships, work, psychological and physical well-being). Finally, participants were asked about their intention to ask for psychological support in the future. All questions regarding the personal experience with the COVID-19 pandemic were addressed after the end of the experimental paradigm to avoid any effect that these questions may have had on performance, thus risking interferences with the effects of the affective prime (especially in the COV condition).

A copy of the experimental paradigm(s), anonymized data, and materials used are openly available on the project's Open Science Framework page (https://osf.io/svb8x). The movie excerpts are also available as Supplemental Information.

## Statistical analysis

Data were analyzed using R (v4.1.2; *R Core Team, 2021*). Participants' percentages of correct responses (thereinafter, Accuracy) and average response times (RTs) were analyzed, separately for ongoing trials and PM cues. RTs were calculated considering correct trials

only and excluding the trials whose RT differed for more than 2.5 standard deviations from the mean. For both ongoing trials and PM cue, two different analyses were computed: first, we performed an analysis of variance (ANOVA) to ascertain the presence of an effect of the variables of interest (primarily an effect of the variable Video). Once this effect was found, we analyzed our results with linear model approaches, to have a direct measure of the involvement of each factor (also considering the responses to the questionnaires) in the explanation of the variability of the results.

In detail, ongoing trials were first analyzed with a mixed-factorial analysis of variance (split-plot ANOVA), with a 2 (Block: baseline *vs.* PM condition; within subjects) × 3 (Video: COV *vs.* DOC *vs.* NEGA; between subjects) design. This analysis explored the presence of an effect of the variable Video, which was found on Accuracy ($F_{(2, 275)} = 4.263$; $p = .015$; $\eta_p^2 = .030$). No effect of this factor was found for RTs, which instead resulted in being influenced by the variable Block ($F_{(1, 275)} = 640.903$; $p < .001$; $\eta_p^2 = .700$). Since an effect of the variable Video was found, performance in ongoing trials was further investigated with the implementation of a linear mixed effect (LME) Model (*Bates, Maechler & Bolker, 2011*). This approach was selected to calculate the effect size of each presented video clip separately, involving the Variable Participant as a random factor (thus comparing performance between blocks within each subject) and the scores in the DASS-21 and the COVID-19-PTSD questionnaires, to determine if any of those values could increase the explanatory power of the model.

Regarding PM cues, Accuracy and RTs were analyzed using an analysis of variance (ANOVA), which indicated no effect of the variable Video. Subsequently, the two metrics were analyzed with a Linear Model approach (LME analysis was not applicable, since PM cues appeared only in the PM condition, thus giving only one measure per participant). In these analyses, the scores in the DASS-21 and the COVID-19-PTSD were also included to evaluate their contribution to the explanation of the variance of the data.

Finally, a correlation analysis was performed between Accuracy and RTs (for both ongoing trials and PM cues) and the responses to the questionnaires. This last analysis was performed using JASP, an open-source statistics program (Version 0.16.3; *JASP Team, 2020*). Since the relationship between the investigated variables could have been nonlinear, we used Spearman's rho index.

## RESULTS

### Ongoing trials
#### *Accuracy*
The split-plot ANOVA did not reveal any significant differences depending on the variable Block ($F_{(1, 275)} = .020$; $p = 0 .887.$; $\eta_p^2 = .0001$). Instead, a medium-size effect was found depending on the variable Video ($F_{(2, 275)} = 3.587$; $p = .029$; $\eta_p^2 = .025$), together with an effect of the interaction between Video and Block ($F_{(2, 275)} = 3.755$; $p = .025$; $\eta_p^2 = .027$). *Post-hoc* analyses revealed no difference between versions (COV, DOC, NEGA) in the baseline, confirming the absence of differences between the three populations before the video was displayed. Instead, in the PM condition participants that were presented
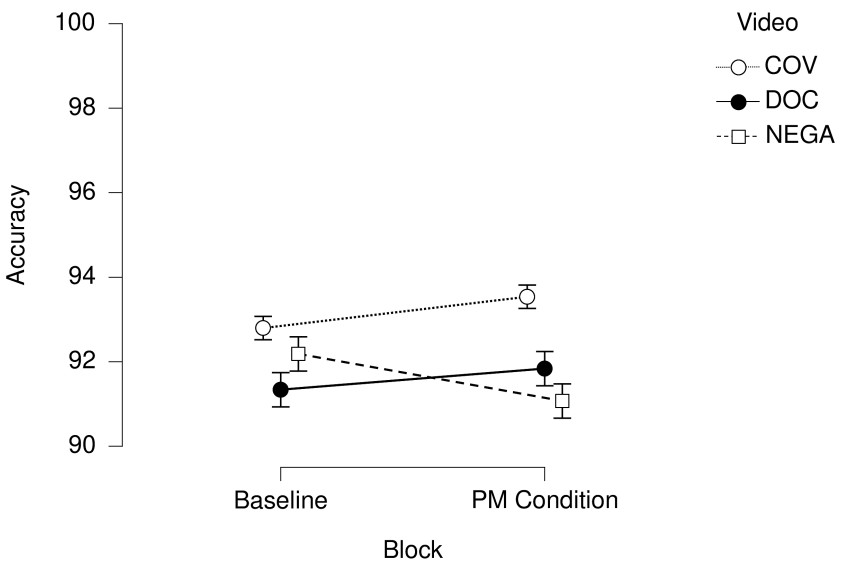

**Figure 1** **Accuracy in the ongoing task.** Correct responses to ongoing trials in the baseline and the PM condition in the three versions of the paradigm. Error bars represent standard Errors.

with COVID-related contents performed significantly better than the ones in the DOC ($t$.ratio $= 2.386$; $p_{\text{Bonf}} = .046$) and the NEGA ($t$.ratio $= 3.323$; $p_{\text{Bonf}} = .003$) versions of the paradigm.

Regarding the PM cost, participants in the COV and DOC versions presented no difference between the baseline and the PM condition in terms of Accuracy. In the NEGA version, on the other hand, a detrimental effect of the video on Accuracy was found (Fig. 1; $t$.ratio $= 2.141$; $p_{\text{Bonf}} = .033$).

In line with these preliminary results, the LME model (Table 3) highlighted an effect of the variable Video on Accuracy, specifically of the video clip whose content referred to the pandemic (COV version; $t$.value $= 2.069$; $p = .039$). The factor Block did not show to significantly affect Accuracy per se, but an effect of its interaction with the variable Video was found, attributable to the PM cost found in the NEGA condition ($t$.value $= -2.187$; $p = .030$). Additionally, the subscales of the DASS-21 and COVID-19-PTSD questionnaires were included in the LME model as fixed factors, to discover if these variables may play a role in the explanation of the dataset. The inclusion of the scores at the DASS-Anxiety subscale resulted in the increase of the model's fitness to the results ($\chi^2 = 8.173$; $p = .004$), indicating a detrimental role of the self-reported levels of anxiety on Accuracy to the ongoing trials (Fig. 2; $t$.value $= -2.859$; $p = .005$).

*Reaction times*

The split-plot ANOVA revealed an effect of the variable Block on RTs, as they were consistently higher (147 ms) in the PM condition compared to the Baseline (Fig. 3). Accordingly, we detected a strong effect depending on the variable Block ($F$ (1, 275) $= 640.903$; $p < .001$; $\eta_p^2 = .700$). On the other hand, no effect depending on the variable Video was found ($F$ (2, 275) $= .096$; $p = .909$; $\eta_p^2 = .0004$), nor on its interaction with the

**Table 3   Summary of the linear mixed-effect model on accuracy in the ongoing task.**

| Predictors | Accuracy | | |
|---|---|---|---|
| | **Estimates** | **CI** | **p. value** |
| (Intercept) | 92.05 | 90.88–93.22 | <0.001 |
| Video (COV) | 1.57 | 0.08–3.07 | 0.039 |
| Video (NEGA) | 0.91 | −0.59–2.40 | 0.233 |
| Block | 0.50 | −0.53–1.53 | 0.340 |
| DASS_Anxiety | −0.10 | −0.17–−0.03 | 0.005 |
| Video (COV) * Block | 0.24 | −1.21–1.69 | 0.743 |
| Video (NEGA) * Block | −1.61 | −3.06–−0.16 | 0.030 |
| Number of Subjects | 278 | | |
| Observations | 556 | | |
| Marginal $R^2$ /Conditional $R^2$ | 0.047/0.552 | | |

**Notes.**
Summary of the effects found with the LME Model on Accuracy in the Ongoing Task. The effects of the "COV" and "NEGA" video clips are compared to the "DOC" condition, in which the neutral video was presented. The effect of Block represents the difference in Accuracy between the baseline and the PM condition.

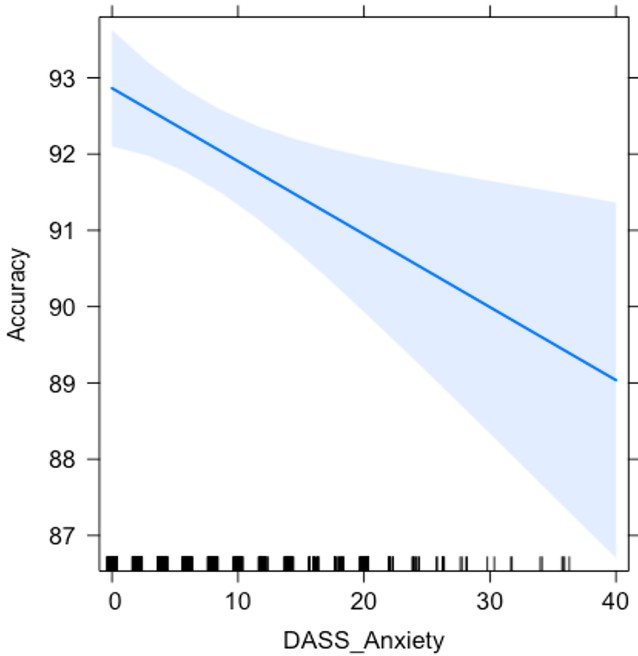

**Figure 2   Relationship between anxiety and accuracy in the ongoing task.** Graph representing the relationship between the scores in the DASS-21 Anxiety subscale and Accuracy to the ongoing trials.

variable Block ($F$ (2, 275) $= .991$; $p = 0.372$; $\eta_p^2 = .01$). Likewise, the LME model showed a consistent effect based on the factor Block ($t$.value $= 13.416$, $p < .001$), but no differences depending on the displayed video clip. The explanatory power of the model built on RTs resulted to be enhanced by the inclusion of the scores in the DASS-Depression subscale as a fixed factor ($\chi^2 = 4.976$; $p = .026$), thus showing that higher levels of self-reported

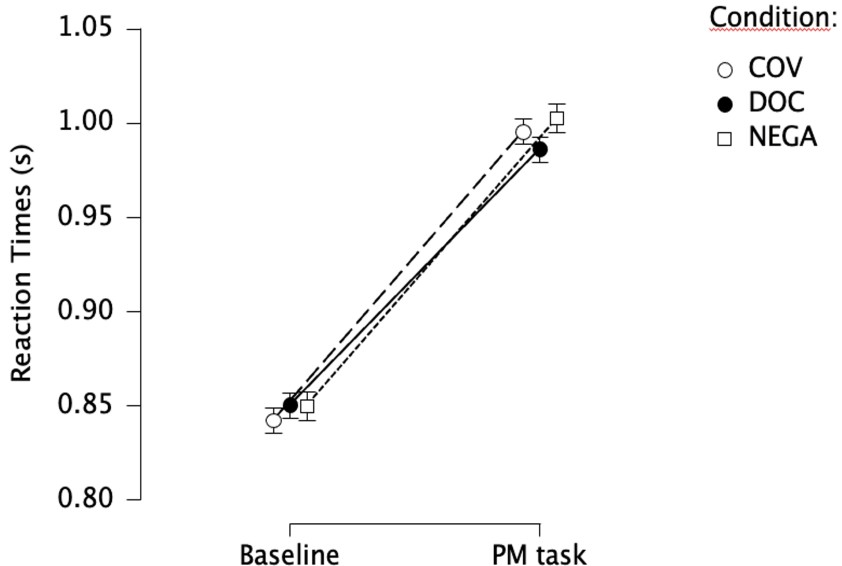

**Figure 3   Reaction times in the ongoing task.** Responses times to ongoing trials in the baseline and the PM condition in the three versions of the paradigm. Error bars represent standard Errors.

**Table 4   Summary of the linear mixed-effect model for reaction times (RTs) in the ongoing task.**

| Predictors | Reaction times | | |
|---|---|---|---|
| | Estimates | CI | *p*.value |
| (Intercept) | 0.83 | 0.80–0.87 | <.001 |
| Video (COV) | −0.01 | −0.06–0.03 | .524 |
| Video (NEGA) | 0.00 | −0.05–0.04 | .897 |
| Block | 0.14 | 0.12–0.16 | <.001 |
| DASS_Depression | 0.00 | 0.00–0.00 | .027 |
| Video (COV) * Block | 0.02 | −0.01–0.05 | .217 |
| Video (NEGA) * Block | 0.02 | −0.01–0.05 | .230 |
| Number of Subjects | 278 | | |
| Observations | 556 | | |
| Marginal $R^2$ /Conditional $R^2$ | 0.212/0.829 | | |

**Notes.**
Summary of the effects found with the LME Model on Reaction Times in the Ongoing Task. The effects of the "COV" and "NEGA" video clips are compared to the "DOC" condition, in which the neutral video was presented. The effect of Block represents the difference in RTs between the baseline and the PM condition.

depression were associated with slower responses ( $t$.value = 2.224, $p$ = .027). The PM cost was consistent among the three versions (DOC: $t$.ratio = −13.416, $p$ < .001; COV: $t$.ratio = −15.245, $p$ < .001; NEGA: $t$.ratio = −15.194, $p$ < .001). A comprehensive list of the factors for the models and their effects is presented in Table 4, and the effect of the DASS-Depression subscale is displayed in Fig. 4.

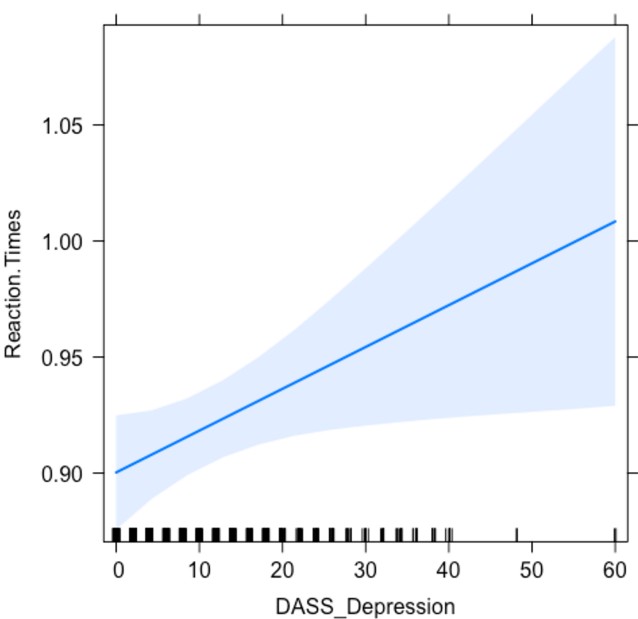

**Figure 4** **Relationship between depression and reaction times (RTs) in the ongoing task.** Graph representing the relationship between the scores in the DASS-21 Depression subscale and RTs to the ongoing trials.

## PM cues
### *Accuracy*

On average, participants correctly detected 5.6 out of 8 (68%) of the presented PM cues. The LM showed no significant differences depending on the displayed video. Furthermore, the model showed to increase his explanatory power when the factor Video was excluded, and only the scores in the DASS-Depression and the DASS-Stress subscales ($F$ (2, 275) = 3.091; $p$ = .047) were included as fixed factors. Interestingly, these two variables were characterized by similar effect sizes, but opposite directions: participants with higher levels of self-reported depression detected PM cues more often ($t$.value = 2.474; $p$ < .014), whereas participants with higher levels of stress detected fewer PM cues ($t$.value = −1.988; $p$ < .048). Table 5 presents in detail the LM and Fig. 5 displays the effects of the two DASS subscales.

### *Reaction times*

PM cues were detected with similar RTs (on average, 1004 ms) independently from the displayed video ($F$ (2, 275) = 2.413; $p$ = .092). None of the variables considered for the LM (Video, scores in the DASS-21, scores in the COVID-19-PTSD scales) affected the explanatory power of the model on PM cues' response times.

## Correlation analysis

Correlation analyses were conducted to identify the presence of associations between performance and the questionnaires participants responded to. Specifically, the interest was in the correlations between Accuracy and RTs (both for PM cues and ongoing trials) and the

**Table 5  Summary of the linear model for accuracy in the PM task.**

| Predictors | Accuracy | | |
| --- | --- | --- | --- |
| | Estimates | CI | p.value |
| (Intercept) | 5.54 | 5.07–6.02 | <.001 |
| DASS_Depression | 0.04 | 0.01–0.08 | .014 |
| DASS_Stress | −0.04 | −0.08–0.00 | .048 |
| Observations | 278 | | |
| $R^2$/$R^2$ adjusted | 0.022/0.015 | | |

**Notes.**
Summary of the effects found with the LME Model on Accuracy in the PM Task. DASS Depression and DASS Stress represent the scores in the respective two subscales of the DASS-21.

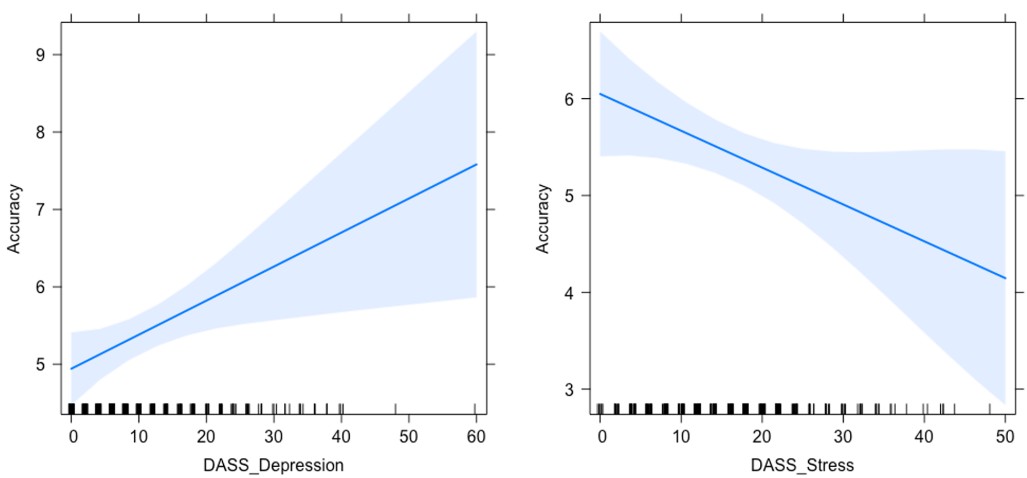

**Figure 5  Relationships between Accuracy to the PM task and Depression (left) and Stress (right).**
Graphs representing the relationship between the scores in the DASS-21 Depression and Stress subscales and Accuracy in PM cue Detection (left and right, respectively).

scores in the DASS-21 and the COVID-19-PTSD scales, and between these questionnaires and the personal experience each participant had with the pandemic. Regarding ongoing trials, Accuracy showed a negative correlation with the scores in the DASS-Anxiety subscale (rho = −.129, $p$ = .002), in line with the results of the LME model. Furthermore, Accuracy was also found to negatively correlate with the scores in the DASS-Depression (rho = −.102, $p$ = .016) and the Intrusion subscales (rho = −.108, $p$ = .011). Regarding response times, a positive correlation was found between the ongoing task's RTs and the scores in the DASS-Depression subscale (rho = .124, $p$ = .003). Accuracy and RTs in the PM task did not correlate with any of the considered variables. The only exception was the tendency between the PM cue's Accuracy and the Intrusion subscale to have a negative correlation. However, this result failed to reach significance (rho = −.108, $p$ = .063).

Significant correlations were found between every subscale of the DASS-21 and the COVID-19-PTSD questionnaires. Specifically, the three subscales of the COVID-19-PTSD questionnaire (Intrusion (I), Avoidance (A), Dysphoric Arousal (DA)) strongly correlated

with the DASS-Anxiety scale (I: rho = .416; A: rho = .355; DA: rho = .351) and, to a lesser extent, also with the scores in the DASS-Stress (I: rho = .334; A: rho = .296; DA: rho = .299) and in the DASS-Depression (I: rho = .262; A: rho = .278; DA: rho = .282) subscales. All results presented a significance of $p < .001$.

The other questions regarding each participant's personal experience with the pandemic also showed interesting associations. In particular, the variable *Fear* (representing the fear of being infected by the COVID-19 virus) showed positive correlations with all the COVID-19-PTSD subscales and, to a lesser extent, also with the subscales of the DASS-21. Furthermore, Fear presented positive correlations with the RTs to the ongoing task, with the perceived impact of the pandemic on physical and psychological health (*Physical_Health* and *Psychological_Health*, respectively) as well as with the intention to ask for psychological support in the future (*Psychological_Help*). Likewise, Psychological_Help showed strong positive correlations with all the subscales of the two questionnaires, as well as with Physical_Health and Psychological_Health. An exhaustive presentation of the correlations between each variable is displayed in Fig. 6.

## DISCUSSION

The present study aimed to explore the impact of COVID-related information on participants' ability to remember to perform an intention in the future (*i.e.,* prospective memory, PM) while simultaneously performing an ongoing activity. To do so, participants were presented with a classical PM paradigm, composed of a baseline and a PM condition, with the addition of an affective prime (a video clip containing either neutral, negative, or COVID-related information) displayed right after the PM condition. Participants who were exposed to the COVID-related video clip (COV condition) resulted to have a significantly better ongoing task performance in the PM condition, compared to the participants who were instead presented with either the neutral or negative video clip. Specifically, accuracy in the ongoing task resulted to be higher after the presentation of the COVID-related prime, whereas no differences were found in terms of performance in the PM task. This result was in contrast with our initial hypothesis, as we expected COVID-related stimuli to have a detrimental effect on PM performance (both in the ongoing and PM tasks). However, our finding resulted to be in line with previous studies that found similar results (improvement of accuracy in the ongoing task, but not in the PM task) after the presentation of stress-inducing stimuli (*Möschl et al., 2017*; *Möschl et al., 2019*; *Nater et al., 2006*; *Walser et al., 2013*). The authors interpreted the enhancement of performance in response to the presentation of acute stressors in light of the *Yerkes & Dodson (1908)*, which posits that intermediate levels of stress can trigger the right amount of arousal to effectively perform the task. In this light, it is possible that the video containing COVID-related information raised a certain amount of stress, not generated by the other affective primes, that had a positive effect on performance in the ongoing task. In line with this hypothesis, other studies found that video clips regarding worrying and real situations induced a sort of "alertness state" and positively affected performance (*Charmandari, Tsigos & Chrousos, 2005*; *Dickerson & Kemeny, 2004*).

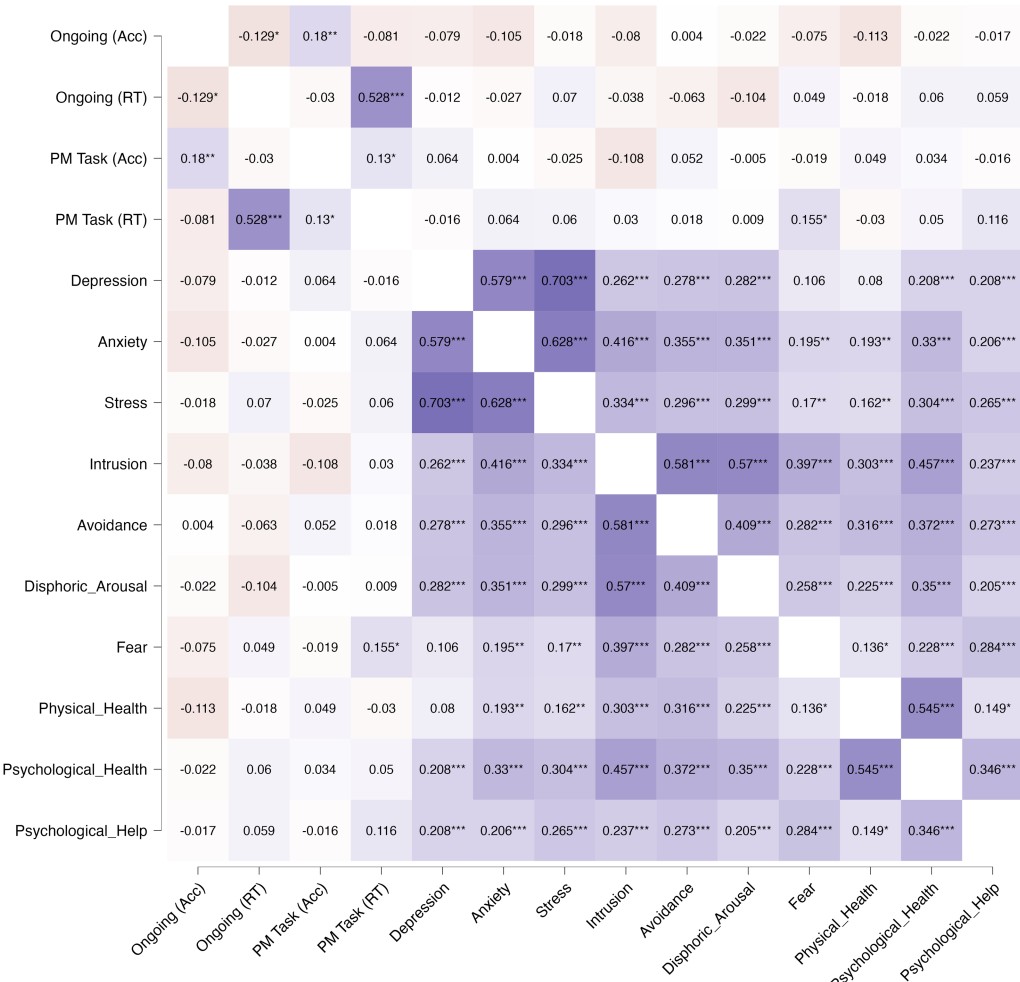

**Figure 6** **Correlation map between performance, questionnaires and personal experience with the pandemic.** Heatmap of the correlations between performance in the Ongoing and the PM tasks (both Accuracy and RTs), the scores in the DASS-21 and COVID-19-PTSD subscales, and the questions regarding (A) How much they fear the COVID-infection (*Fear*), (B) the impact of the pandemic on their physical and mental health (*Physical_Health and Psychological_Health*), and (C) the possibility of seeking psychological support in the future (*Psychological_Help*). * $p < .05$, ** $p < .01$, *** $p < .001$.

The robustness of the association between accuracy in the ongoing task and the displayed video is increased by the fact that the linear mixed effect (LME) computed to investigate which factor had a strong influence on performance showed a significant effect of the video clip displaying COVID-related contents. Additionally, accuracy to the same task (LDT) was identical between the three populations in the baseline condition, thus proving the absence of significant differences before the presentation of the video clips.

In the condition presenting a standard negative video clip (NEGA), performance in the ongoing and the PM tasks did not differ significantly from the condition presenting the neutral prime (DOC), neither in terms of accuracy nor response times. Intriguingly, however, the NEGA condition was associated with a significant decrease in ongoing task

accuracy between the baseline and the PM condition. The detrimental effect of a negative prime is in line with a meta-analysis on the influence of emotional cues on PM (*Hostler, Wood & Armitage, 2018*). The presence of this effect selectively in the NEGA condition, but not in the COV one, further distinguishes the effects of the COVID-related video clip from the effects of more classical negative primes.

Along with the effects of the video clips, the analysis also revealed that the scores in the Anxiety subscale of the DASS-21 increased the explanatory power of the LME Model computed on the ongoing trials' accuracy. In other words, participants' accuracy was associated with anxiety, as higher levels of state anxiety were paired with lower performance in the ongoing task. This result is in line with several studies that found detrimental effects of anxiety on PM (*Arnold, Bayen & Böhm, 2015*; *Harris, 1999*; *Kliegel & Jäger, 2006*).

Regarding reaction times (RTs), no difference was found depending on the presented video clip, as RTs were overlapping between versions in both the Baseline (∼850 ms) and the PM conditions (∼1000 ms). On the other hand, a consistent effect depending on block (increased RTs to ongoing trials in the PM condition compared to the Baseline, attributable to the addition of the prospective task) was found across conditions. This difference, known in the literature as the PM cost, is considered to reflect strategic monitoring processing (*Anderson, Strube & Mcdaniel, 2019*). Accordingly, the consistent presence of PM costs independently from the displayed video clip can be interpreted as the result of the increased active monitoring required to detect the non-focal, non-salient PM cues among ongoing trials, a process that appears to be unaffected by the valence and arousal of the displayed video clip.

Participants' scores in the Depression subscale of the DASS-21 were found to increase the explanatory power of the LME model on their RTs to the ongoing task. Specifically, participants with higher scores in the Depression subscale tended to respond slower. These results are in line with several cross-sectional studies that found people with more severe depressive symptoms to perform slower in several cognitive tasks (*Austin, Mitchell & Goodwin, 2001*; *McDermott & Ebmeier, 2009*; *Gualtieri & Morgan, 2008*).

Regarding PM cues, their detection wasn't affected by the presented video clip (neither in terms of accuracy nor RTs). Regarding the COV condition, this result is in line with the hypothesis of this stimulus working as an acute stressor rather than as a standard negative prime, since the same pattern of results was found in previous studies for stress-inducing primes (*Möschl et al., 2017*; *Möschl et al., 2019*). In the NEGA condition, instead, interpretation is less straightforward. A possible explanation relies on the results of *Hostler, Wood & Armitage (2018)*, who found that the impact of negative stimuli on PM cue detection requires the associated negative valence to be present both during the encoding and retrieval phases. Thus, the absence of effects of the video clip may rely on the fact that the presentation of an affective prime (either COVID-related or negative) during the encoding stage, despite significantly impairing ongoing monitoring processes, was not strong or long-lasting enough to affect PM retrieval.

The linear model (LM) that resulted to better explain the variance in the PM cue accuracy was the one selectively involving the scores in the Depression and Stress subscales of the DASS-21 questionnaire. In detail, participants with higher scores in the Depression subscale

were found to detect PM cues more often, whereas higher scores in the Stress subscale were associated with lower accuracy. The association between depressive symptoms and enhanced accuracy in the PM task can be interpreted in the light of the affective-regulation-of-processing model (*Storbeck & Clore, 2008*), which posits that negative affective states lead to more item-specific processing. In line also with *Rummel et al. (2012)*, this result may indicate that experiencing more negative states can result in improved item-specific processing, thus enhancing the detection of specific features of the stimuli, such as specific syllables (the PM cues). Regarding the relationship between PM cue detection and stress, results in the literature are mixed and clouded by methodological differences across studies (see *Piefke & Glienke, 2017* for a review). In general, however, studies that collected self-report measures of stress reported a pattern similar to the one in this study, that is, a tendency of accuracy to the PM task to decrease with increased levels of stress (*Ihle et al., 2012*).

The involvement of different questionnaires on participants' well-being and experience with the pandemic not only contributed to the explanation of the results in the PM paradigm but represented a fruitful resource of information per se. Namely, the employment of the Italian adaptation of the DASS-21 scale (*Bottesi et al., 2015*) allowed us to gain a robust measure of depression, anxiety, and stress symptoms in the 7 days preceding the acquisition. The three subscales selected from the COVID-19-PTSD questionnaire provided insights into the widespread presence of intrusive thoughts (Intrusion), tendencies to avoid thoughts or reminders (Avoidance), and sleep or concentration difficulties (Dysphoric Arousal) related to the pandemic experience. Finally, the questions regarding participants' personal experience with COVID-19 provided useful data to identify individual factors that may have played a role in the effects of the affective prime on PM performance.

Responses to the DASS-21 scale revealed important information regarding the role of anxiety, stress, and depression levels on PM, affecting performance in the ongoing task. Specifically, higher anxiety scores were associated with lower accuracy, whereas higher depression scores were paired with longer RTs. The COVID-19-PTSD questionnaire and the other questions did not demonstrate an analogous explanatory power on performance. Nonetheless, these data helped to shed light on more qualitative aspects of each participant's experience with COVID-19. In detail, the correlational analysis between the DASS-21 and the COVID-19-PTSD subscales showed strong associations between PTSD-like symptoms derived from the pandemic experience and depression, stress, and state anxiety. Anxiety levels, in particular, showed the highest correlation with the three PTSD symptoms scales, corroborating the hypothesis of its importance in the explanation of the impact that COVID-related stimuli may have had on participant's well-being and everyday functioning. The COVID-19-PTSD subscales also showed correlations with both the fear of the infection and the perceived impact of the pandemic on psychological health. This was particularly true for the Intrusion subscale, suggesting that participants who experienced this type of intrusive thoughts more often tended to be more worried about the infection and perceived their mental health as more affected. The intention to ask for psychological support in the future was weakly, but significantly, correlated with all of the DASS-21 and COVID-19-PTSD subscales, indicating that, regardless of the symptoms, more than half of

the participants (176 out of 278 participants, 63%), were considering psychological support to face the aftermath of the pandemic at the time of the acquisition (more than one year after the general lockdown in Italy), a situation that should not remain underestimated.

## CONCLUSIONS

This study explored the effects of exposure to COVID-related information on PM. To do so, we applied an affective priming paradigm in which different video clips were used as affective primes, displaying either neutral, negative, or COVID-related content. This paradigm was used to mimic the everyday experience of continuous mediatic exposure to the pandemic, exploring the hypothesis that such information may interfere with PM. Furthermore, we collected data to investigate the possible presence of symptoms of depression, anxiety, and PTSD in the participants, along with information on their personal experiences with the pandemic. The exposure to a COVID-related video clip right before the beginning of the PM condition resulted in improved accuracy in the ongoing task, compared to the conditions in which other video clips were displayed. This result could be explained by previous studies that revealed that acute stressors (*e.g.*, reminders of negative situations participants were currently experiencing) might enhance accuracy in the PM condition, selectively to ongoing trials. As further confirmation that several features influenced ongoing task processing, anxiety levels showed a detrimental effect on accuracy, as higher scores in the DASS-Anxiety subscale were associated with lower performance. Ongoing task's RTs, on the other hand, did not differ depending on the displayed video clip but were associated with the scores in the DASS-Depression subscale, as higher depressive levels were associated with longer response times. Regarding the PM task, the displayed video clip did not affect PM cue detection, neither in terms of accuracy nor RTs. The absence of an impact of the COVID-related video on the PM task accounts as additional evidence of its processing as an acute stressor, as the same patterns were found in previous studies on this type of stimuli. However, it may also suggest that the selected prime was not strong enough to trigger any interference, also considering the small number of trials that a PM task must be composed of. The linear model that better explained the accuracy in the PM task resulted to be the one involving participants' stress and depression scores in the respective DASS-21 subscales. Specifically, higher levels of depression were associated with higher accuracy (in line with the hypothesis of enhanced item-specific processing) whereas higher levels of stress were paired with lower accuracy.

The presence of PTSD symptoms associated with the experience of the pandemic (collected through the administration of the COVID-19-PSTD scale) did not affect performance in the PM paradigm. Critically, however, the presence of such symptoms (particularly intrusive thoughts) was associated with anxiety levels, fear of the infection, and the perceived impact of the pandemic on both psychological and physical health. Further investigations are not only required, but necessary, to provide new insights on the effects of the emotional states induced by COVID-related information on cognition. Future studies will contribute to gain a better understanding of the aftermath of the pandemic, oriented to the creation of coping strategies to face the consequences of long exposure to this peculiar and stressful situation.

## ACKNOWLEDGEMENTS

We thank Lavinia Conoscenti and Elena Piazzalunga for their contribution to data collection and analyses. Portions of the text presented in this manuscript were previously published as part of a preprint (*Vicentin et al., 2023*)

### Funding

The authors received no funding for this work.

### Competing Interests

The authors declare there are no competing interests.

### Author Contributions

- Stefano Vicentin conceived and designed the experiments, performed the experiments, analyzed the data, prepared figures and/or tables, and approved the final draft.
- Giovanni Cantarella performed the experiments, analyzed the data, prepared figures and/or tables, and approved the final draft.
- Giorgia Cona conceived and designed the experiments, authored or reviewed drafts of the article, and approved the final draft.
- Patrizia Bisiacchi conceived and designed the experiments, authored or reviewed drafts of the article, and approved the final draft.

### Human Ethics

The following information was supplied relating to ethical approvals (*i.e.*, approving body and any reference numbers):

The University of Padua granted Ethical approval to carry out the study within its facilities (Protocol Number: 3931).

### Data Availability

A copy of the experimental paradigm(s), anonymized data and materials used are available at the Open Science Framework: Vicentin, Stefano. 2023. "THAI—The Impact of COVID-Related Information on Prospective Memory". OSF. June 26. osf.io/svb8x.

### Supplemental Information

Supplemental information for this article can be found online at http://dx.doi.org/10.7717/peerj.16389#supplemental-information.

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
