# Peer review of "Thinking about it: the impact of COVID-19-related stimuli on prospective memory"

_PeerJ, doi:10.7717/peerj.16389_

## Round 0.1 · original submission · Major Revisions

This is a straightforward action letter to write. I won’t comment on the reviewers’ critiques in too much detail, as I feel that they are both constructively critical and quite clear. I’ll just mention a few of the major points raised by the reviewers.

Rev1 notes that the writing is largely good, but in need of some cleaning up. S/he also notes positive aspects about the research question, design, and analysis. The reviewer goes on to note, however, that the Discussion section is in need of more nuance with respect to how it describes research findings, and the comments raised here will definitely need to be addressed because the reviewer is suggesting a slightly different interpretation of the results than you offer in the manuscript currently. Other more minor comments are offered, but I’ll just let you read those for yourself.

Rev2 offers a review that is very much in line with Rev1’s, and is perhaps a bit more critical. This reviewer notes more difficulty with the language, and I must say that that was my reading of the manuscript as well. This refers not just to the wording, but the general organization and manner in which the sections connected to one another. This reviewer also notes concerns with how the introduction aligns with the motivation of your design, and the degree to which the results are transparently discussed (i.e., the experiment does not appear to have worked out as the original motivations would have predicted). Other comments are offered that I will leave you to read for yourselves.

Overall, there is a fair bit of work to do. This is a potentially interesting study, but the writing clearly needs considerable attention. And I’m quite seriously concerned about the Discussion in particular, seeing that more nuance is needed (as Rev1 points out) and seeing that the initial goals of the study do not seem well aligned with the discussion of what the results mean (as Rev2 points out). I generally try not to bother reviewers with a second review when I can help it, but it seems likely that I will need to send this paper back to one or both for another look at your revised manuscript.

·

Basic reporting

• Language and writing were good, however, there are few grammatical errors throughout the manuscript (no impact on content comprehension though).
• The introduction and background literature was relevant and appropriate citations were included.
• Figures are relevant and well-labeled, but the quality of the image is low-resolution.
• Raw data is provided and accessible.

Experimental design

• To my knowledge, the experimental design involves original primary research that bridges a gap in the PM literature.
• The research question, and consequent hypothesis, were clear and well defined.
• Methodology involves classical paradigms that were well-established in the literature.

Validity of the findings

• The raw data files are provided, and the statistical analyses employed are appropriate for the data.
• However, conclusions from results need to be qualified a bit more, as the results seem more nuanced than what is described in Abstract, Discussion, and Conclusion sections. Here is my explanation:

Overall, this was an interesting investigation with clear hypothesis and strong experimental design. Throughout, the manuscript does do a great job at reviewing previous work on the area, and the authors consistently connect the findings to prior published studies. However, I believe that the finding that Covid-related information positively impacts performance in the PM task is more nuanced than how it is described in the Discussion section (lines 455-460). Particularly, the video manipulation seems to only impact accuracy for the ongoing task, but not detection of PM cues (lines 405-407). Typically, in PM studies, participants divide their attention between the ongoing task (lexical decision) and the PM task (monitoring for and detecting PM cues), with performance improving in one task as it decreases in the other, depending on what is prioritized by participants (so-called “attention-allocation policy” employed by participant; Marsh, Hicks, & Cook, 2005). In other words, participant tend to focus attention resources on one task only, and they can switch back-and-forth tasks as the experiment progresses. However, because ongoing task RTs were also not impacted by the video manipulation (lines 390-393), it seems that participants were focusing primarily on the lexical decision task, and not on monitoring for PM cues (or at least that the video manipulation did not influence the degree of monitoring engaged by participants in the study, as indicated in lines 396-397).

As currently described in the manuscript, both in the Abstract and Conclusion, the authors claim that the “…Covid-related clip performed significantly better in the PM condition than participants in the other conditions…” (lines 36-36) and that “…exposure to a Covid-related video clip right before the beginning of the PM task resulted to improve accuracy in the PM condition…” (lines 567-569). I believe these, and similar statements in the Discussion, may be a little bit misleading as the Covid-related video did not facilitate detection of PM cues, nor it increased monitoring for such cues. I invite the authors to qualify their conclusions by highlighting that, when they talk about “accuracy” or “performance,” they are indeed referring to accuracy/performance in the ongoing task of the experiment, not the PM task. This will help readers to fully understand the insights of the current study, as well as its implications. Particularly, that the PM process at play here (monitoring and cue detection) seems to be influenced by general anxiety-, stress-, and depression-related states (as captured by the DASS), rather than by Covid-specific information. If any, the impact of Covid-19 related stimuli on prospective memory seems to be minimal here.

Additional comments

Additional Minor comments/queries:

On line 250, it is indicated that the “Cough” excerpt was chosen because it presented valence and arousal levels similar to the negative video. Descriptive stats are provided in Table 1, however, were there any specific analyses conducted (i.e., t-test?) to demonstrate that these vides are indeed similar to one another in terms of valence and arousal?

Across dependent measures, the manuscript describes a series of repeated measures were adopted prior to the LME models. I believe these are actually “mixed” model ANOVAs, as the Video version is a between-groups variable.

On line 346, the authors describe the results of an ANOVA with a F-ratio of 4.263. However, I am unsure here what was being analyzed here (Accuracy or RTs?) as it says that, “Once an effect of Video was found, RTs and Accuracy were investigated, separately…” So I am not sure what the dependent measures is reflected in that F-ratio. Do nonsignificant effects of Video on dependent measures such RTs and PM Cues precludes the LME model?

·

Basic reporting

My concern here is that the way some things are worded, especially in the Method and Results sections, makes the text difficult to understand in places. First, it was difficult to understand the method and results for the video clip validation. And second, several things made it difficult to understand the results of the main experiment. This could perhaps be improved by more exposition about why the different analytic techniques were used, as well as why the different dependent measures (both accuracy and latency for both prospective memory and lexical decision) were used. The subsection titled Statistical Analysis was a bit confusing in terms of its relation to the subsequent subsections. It was also confusing that the authors used the term PM task to refer to performance on the ongoing lexical decision task when prospective memory was embedded (as opposed to when prospective memory was not embedded); typically PM task is used to refer to the prospective memory task itself, not the ongoing task condition in which the prospective memory task is embedded.

Experimental design

My concern here is that I am not confident that the current work fits within the Aims and Scope of the journal. The current research fits most clearly in the Social Sciences, and although there is some relevance to Medical or Health science (e.g., in terms of mental health consequences of the covid-19 pandemic), I am not sure it is sufficient. The authors review some of the literature on the mental health aspects of the pandemic, and their results show some correlations between self-reported aspects of mental health and reactions to the pandemic, but there was no evidence that prospective memory is affected by exposure to covid-relevant stimuli or that it is related to self-reported mental health (other than a possible speed-accuracy tradeoff mentioned below).

Validity of the findings

My concern here is that there is an inconsistency between the Introduction and the Results/Conclusions. The experiment is motivated as a study of the effect of type of video on prospective memory but the results showed no effect here. Although the authors do not disguise this, they also never acknowledge that the experiment did not really work out in terms of why it was conducted. Specifically, the effect of type of video was on lexical decision (i.e., the ongoing task in which the prospective memory task was embedded) and not on prospective memory.

Additional comments

I think it could be useful to provide some explanation for why the different statistical approaches were used. I have never used a mixed effects model and I suspect it would be useful to others like me to explain why it was used in the current situation (e.g., why just a linear model does not suffice). Also, to the extent that an analysis of variance is just a type of linear model, it would be good to explain why both approaches were used.

If the authors submit a revision, it would be nice, if possible, to provide an integrated explanation for the entire set of results, being sure to make clear that prospective memory (the primary dependent measure) was not affected by the type of video (the primary manipulation). Along the lines of an integrated explanation, it could be helpful to try to reconcile the fact that (1) although the covid video (an “acute stressor”) improved ongoing task performance, self-reported anxiety correlated negatively with ongoing task performance (e.g., these could be viewed as contradictory patterns); and (2) self-reported depression correlated positively with prospective memory accuracy (more depressed people were more accurate) but it also correlated positively with prospective memory reaction time (more depressed people were slower), so a speed-accuracy tradeoff seems to be the most parsimonious explanation.

It could be helpful to include more information about why ongoing task performance was analyzed. The authors mention that spontaneous retrieval and strategic monitoring are two ways that a prospective memory task could be accomplished, but it could be useful to be more explicit about how this can be revealed through ongoing task performance. Specifically, lexical decision impairment (either less accurate or slower performance) when there is an embedded prospective memory task, compared to when there is not, is typically viewed as evidence for strategic monitoring. An absence of a difference is typically viewed as evidence for spontaneous retrieval. The current experiment does not allow straightforward interpretation of any such effect, because lexical decision without prospective memory was always performed before lexical decision with prospective memory. Nonetheless, including this information in the Introduction could provide context for why ongoing task performance was analyzed.

---

## Round 0.2 · Major Revisions

Thank you for your revisions to the original manuscript. It is much improved, but there is still a little bit of work to be done. Please see the reviewer's comments, and be sure to address each of them.

·

Basic reporting

• Language and writing was improved.
• The introduction and background literature was relevant and appropriate citations were included.
• Figures are relevant and well-labeled, but the quality of the image is low-resolution.
• Raw data is provided and accessible.

I appreciate the researchers’ responses to our comments and feedback, and the tracked-changes version of the manuscript was very helpful to identify the new edits made. Overall, there has been an improvement in the fluency of the manuscript, as many grammatical issues have been resolved.

Experimental design

• To my knowledge, the experimental design involves original primary research that bridges a gap in the PM literature.
• The research question, and consequent hypothesis, were clear and well defined.
• Methodology involves classical paradigms that were well-established in the literature.

Validity of the findings

I believe that the conclusions are now more aligned with what the result reflects. Particularly, I appreciate the discussion on lines 506-518, where it is highlighted that the results are in contrast with the original hypothesis, yet can be accounted by other explanations (i.e., acute-stress) already established in the literature (Möschl et al., 2017, 2019; Nater et al., 2006; Walser et al., 2016). I agree with such an interpretation of the results, as it seems to me that the acute stress from the Covid video somehow “deactivates” the PM intention, allowing participants to focus attention resources on the ongoing task. However, I think this interpretation of the data can still be qualified better in certain areas of the manuscript. For instance, on lines 514-516, where it is stated “In this light, it is possible that the video containing COVID-related information raised a certain amount of stress, not generated by the other affective primes, that had a positive effect on performance,” I would explicitly add the words “ongoing task” at the end to highlight that the positive effects are on ongoing task performance, not PM performance. Similarly, in the Abstact, the very last line reads “This counterintuitive finding seems to suggest that COVID-related stimuli are processed as “acute stressors” rather than negative stimuli, thus inducing a state of increased alertness and responsivity” I would again indicate that the “increased alertness and responsivity” is for ongoing task stimuli only, not for PM stimuli. In other words, such an increase in alertness is not general or “global” one, but one that is selectively focused on ongoing task stimuli, not just any stimuli in general. Lastly, on lines 626-627, it is worth noting that “PM processing” is not what seems to be affected here (as there were no changes based on video/anxiety on PM costs or PM cue detection). Instead, it is the “ongoing task processing” that seems to be influenced by stress and anxiety.

I recognize these edits may be seen as “neat-picky,” but I truly believe these are key areas of the manuscript where misinterpretation of the findings can easily arise, and thus explicit clarity is required. Additionally, by removing any ambiguity from these sections, the novelty of findings of the current investigation are incidentally highlighted as well.

Additional comments

Minor Comments/Queries:

I appreciate the researchers’ willingness to conduct the independent-samples t-test for the video validation section (lines 319-321). However, in the interest of modeling good transparency practices, I encourage the researchers to also report the results of the t-test within the text (t-score, df, p-value, etc…) even if the difference between the two videos was nonsignificant.

---

## Round 0.3 · accepted · Accept

Thank you for your revisions, and for addressing the reviewer's final concerns. This manuscript is ready for publication.